# Shotgun proteomic profiling of dormant, 'non-culturable' *Mycobacterium tuberculosis*

**Vadim Nikitushkin**[1]*, **Margarita Shleeva**[1], **Dmitry Loginov**[2,3,4], **Filip Dyčka F.**[2]*, **Jan Sterba**[2], **Arseny Kaprelyants**[1]

**1** A.N. Bach Institute of Biochemistry, Federal Research Centre 'Fundamentals of Biotechnology' of the Russian Academy of Sciences, Moscow, Russia, **2** Faculty of Science, University of South Bohemia, Branišovská, Czech Republic, **3** BioCeV—Institute of Microbiology of the CAS, Vestec, Czech Republic, **4** Orekhovich Institute of Biomedical Chemistry, Moscow, Russia

* vadimchemist@gmail.com (VN); fdycka@prf.jcu.cz (FDF)

**Data Availability Statement:** Data are available via ProteomeXchange with identifier PXD028849.

**Funding:** This work was funded by the Russian Science Foundation – Grant 19-15-00324

## Abstract

Dormant cells of *Mycobacterium tuberculosis*, in addition to low metabolic activity and a high level of drug resistance, are characterized by 'non-culturability'–a specific reversible state of the inability of the cells to grow on solid media. The biochemical characterization of this physiological state of the pathogen is only superficial, pending clarification of the metabolic processes that may exist in such cells. In this study, applying LC-MS proteomic profiling, we report the analysis of proteins accumulated in dormant, 'non-culturable' *M. tuberculosis* cells in an *in vitro* model of self-acidification of mycobacteria in the post-stationary phase, simulating the *in vivo* persistence conditions—the raw data are available via ProteomeXchange with identifier PXD028849. This approach revealed the preservation of 1379 proteins in cells after 5 months of storage in dormancy; among them, 468 proteins were statistically different from those in the actively growing cells and bore a positive fold change (FC). Differential analysis revealed the proteins of the pH-dependent regulatory system PhoP and allowed the reconstruction of the reactions of central carbon/glycerol metabolism, as well as revealing the salvaged pathways of mycothiol and UMP biosynthesis, establishing the cohort of survival enzymes of dormancy. The annotated pathways mirror the adaptation of the mycobacterial metabolic machinery to life within lipid-rich macrophages: especially the involvement of the methyl citrate and glyoxylate pathways. Thus, the current *in vitro* model of *M. tuberculosis* self-acidification reflects the biochemical adaptation of these bacteria to persistence *in vivo*. Comparative analysis with published proteins displaying antigenic properties makes it possible to distinguish immunoreactive proteins among the proteins bearing a positive FC in dormancy, which may include specific antigens of latent tuberculosis. Additionally, the biotransformatory enzymes (oxidoreductases and hydrolases) capable of prodrug activation and stored up in the dormant state were annotated. These findings may potentially lead to the discovery of immunodiagnostic tests for early latent tuberculosis and trigger the discovery of efficient drugs/prodrugs with potency against non-replicating, dormant populations of mycobacteria.

(microbiological experiments: cells cultivation and quantification, extracts preparation, data analysis and manuscript preeparation) and by the European Regional Development Fund-Project "Mechanisms and dynamics of macromolecular complexes: from single molecules to cells" (No. CZ.02.1.01/0.0/0.0/ 15_003/0000441) – LC-MS profiling of cells' extracts and proteins annotation. The funders had no role in study design, data collection and analysis, decision to publish, or preparation of the manuscript.

**Competing interests:** The authors have declared that no competing interests exist.

# Introduction

Tuberculosis (TB), caused by the pathogenic microorganism *Mycobacterium tuberculosis* (*Mtb*), currently surpasses HIV as the most common cause of mortality from an infectious disease worldwide (WHO global report).

In the course of infection establishment, *Mtb* passes through a series of intra- and extracellular locations–from alveolar macrophages to extracellular lesions, being exposed to various stress conditions including lowered pH values, nutrient and cofactor limitation, and peroxynitrite stress [1]. Nonetheless, due to its unique metabolic plasticity, the bacterium survives these attacks, gradually transiting into a state of low metabolic activity–dormancy, associated with treatment failure for latent TB infection [2,3]. Bacterial internalization within the host is commanded by virulence factors on the one hand and reprogrammed metabolic networks, balancing the anabolic and catabolic processes with sufficient ATP generation to support survival [4–7].

Disseminated in the host, dormant bacilli may persist life-long; however, approx. 10% of latently infected TB carriers develop active disease [3]. Dormant bacteria obtained from *in vitro* models are characterized by altered morphological traits (such as a thickened cell wall or cell-size diminishment), or resistance to antibiotics [8,9]. However, the cells obtained in such 'quick' models do not reflect mycobacterial physiology *in vivo*, where bacilli transit to a state of 'non-culturability'–a specific reverse state of inability to grow on solid media, and dependent on a reactivation procedure in selective liquid medium [10].

Whether metabolic pathways are active in such a state of 'non-culturable' dormancy of *Mtb* is not known and any knowledge in this respect will be valuable for understanding the survival mechanism and for selection and finding a remedy against latent TB. To elucidate such mechanisms, knowledge of the enzymes preserved in long-term stored dormant cells is essential.

In the case of dormant cells that are metabolically inert, and therefore tolerant to common antibiotics, there is an attractive idea to consider ubiquitous enzymatic processes stored in dormancy as targets for the conversion of prodrug compounds into substances with unspecific antibacterial activity (like reactive oxygen and nitrogen species or antimetabolites) [11,12].

Therefore, analysis of the enzymes stored up in dormancy may be prospective in terms of further development of follow-up prodrugs.

In the current study we analysed the proteomic profiles of *Mtb* cells in the dormant state (after 5 months of storage) in an *in vitro* dormancy model developed earlier [13]. Essentially, the transition to dormant state of *M. tuberculosis* in this model is based on gradual self acidification of the medium in prolonged stationary phase. It is known that during the initial infection, bacteria are engulfed mainly by alveolar macrophages, which serve as the main niche for their growth during this period. Under these conditions stress factors such as nitric oxide, hydrolytic enzymes of lysosomes, reactive oxygen species, and low pH values act on *M. tuberculosis* cells. *M. tuberculosis* cells are able to adapt to stress conditions inside the macrophage, namely, to the effects of slightly acidic pH values (5.8–6.5) what leads to appearance of non-culturable cells insight of macrophage [14]. Based on the fact that a decrease in pH is the main factor limiting the growth of *Mtb* in macrophages [15] and one of the first stressors that a pathogen encounters when it enters the host organism, one can expect the formation of dormant forms of mycobacteria under conditions of a gradual decrease in the pH value of the medium. This assumption was experimentally confirmed [13]. The dormant forms of mycobacteria obtained under these conditions have the main characteristics of mycobacteria ("non-culturability" on standard media, resistance to antibiotics, altered morphology) detected in period of latent tuberculosis. It is important that when mice were infected with dormant "non-culturable" mycobacteria obtained under conditions of gradient acidification in the stationary phase,

the tuberculous process developed after a year and a half in mice upon reaching the early stages of aging [16,17].

Apparently, the weakening of the immune system of mice leads to the reactivation of dormant forms of *M. tuberculosis*, which is similar to the situation observed in latent tuberculosis in humans. Thus, these specialized dormant forms of mycobacteria, obtained in this *in vitro* model retain their infectious potential for a long time what is characteristic for latent TB. Overall, the characteristic properties of"non-culturable" cells and their behavior *in vivo* are in favor of applicability of this approach for modelling latent TB.

In our recent publication, using 2D electrophoresis-based proteomics, we found that 1-year-old dormant, 'non-culturable' cells maintain a significant amount of diverse proteins, including enzymes belonging to different biochemical pathways [18]. However, this approach has evident limitations connected, firstly, with only partial coverage of the whole *Mtb* proteome and, secondly, with the inability to quantitatively estimate the differences in the abundancy of particular proteins between the groups of cells analysed (active vs dormant cells). LC-MS-based proteomic technologies are ideally suited for such comparative analysis, evading the above-mentioned problems of a 2D approach.

## Materials and methods

### Bacterial strains, growth media and culture conditions

Inoculum was initially grown from frozen stock stored at −70˚C. *Mtb* strain H37Rv was grown for 8 days (up to OD600 = 2.0) in Middlebrook 7H9 liquid medium (HiMedia, India) supplemented with 0.05% Tween 80 and 10% ADC (albumin, glucose, catalase) growth supplement (HiMedia, India). 200 μl of this culture was added to 50 ml unmodified Sauton medium, containing (per litre): $KH_2PO_4$, 0.5 g; $MgSO_4 \cdot 7H_2O$, 1.4 g; L-asparagine, 4 g; glycerol, 60 ml; ferric ammonium citrate, 0.05 g; sodium citrate, 2 g; 1% $ZnSO_4 \cdot 7H_2O$, 0.1 ml; $H_2O$, to l L; pH 7.0 (adjusted with 1 M NaOH) and supplemented with 0.05% Tween 80 and 10% ADC and this culture was grown for 10 days at 37˚C with agitation (200 rpm). The culture, grown in this medium, served as an inoculum that was added to modified medium to reach the concentration ~ $10^5$–$10^6$ cells per ml (ca. 2 ml / 200 ml). The composition of the modified Sauton medium is following: $KH_2PO_4$, 0.5 g; $MgSO_4 \cdot 7H_2O$, 1.4 g; L-asparagine, 4 g; glycerol, 2 mL; ferric ammonium citrate, 0.05 g; citric acid, 2 g; 1% $ZnSO_4 \cdot 7H_2O$, 0.1 mL; pH 6.0–6.2 (adjusted with 1 M NaOH). Moreover this modification includes a supplement: 0.5% BSA (Cohn Analog, Sigma), 0.025% tyloxapol and 5% glucose. Cultures were incubated in 500 mL flasks containing 200 mL of modified Sauton medium at 37˚C with shaking at 200 rpm (Innova, New Brunswick) for 30–50 days, and pH values were periodically measured. In log phase, the pH of the culture reached 7.5–8.0 and then decreased in the stationary phase. When the medium in post-stationary phase *Mtb* cultures reached pH 6.0–6.2 (after 30–45 days of incubation), cultures (50 mL) were transferred to 50 mL plastic tightly capped tubes and kept under static conditions, without agitation, at room temperature for up to 5 months post-inoculation. At the time of transfer, 2-(N-morpholino)-ethane sulphonic acid (MES) was added at a final concentration of 100 mM to dormant cell cultures to prevent fast acidification of the spent medium during long-term storage.

### Evaluation of cell viability by CFU and MPN assays

Bacterial samples (age of active mycobacteria—10 days, dormant—5 months) were serially diluted in fresh Sauton medium supplemented with 0.05% tyloxapol; three replicates of each sample from each dilution were spotted on agar plates (1.5%) with Middlebrook 7H9 medium

(HiMedia, India), supplemented with 10% (v/v) ADC (HiMedia, India). Plates were incubated for 30 days at 37˚C, after which the number of CFU was counted.

The most probable number (MPN) assay is a statistical approach based on diluting a bacterial suspension until reaching the point when no cell is transited into any well. The growth pattern in three tubes is compared to the published statistical tables in order to estimate the total number of bacteria (either viable or dormant) present in the sample [19].

The MPN assay was carried out in 48-well plastic plates filled with 0.9 mL of special medium for the most effective reactivation of dormant *Mtb* cells. This medium contains 3.25 g of nutrient broth (HiMedia, India) dissolved in 1 L of a mixture of Sauton medium (0.5 g $KH_2PO_4$; 1.4 g $MgSO_4 \cdot 7H_2O$; 4 g L-asparagine; 0.05 g ferric ammonium citrate; 2 g sodium citrate; 0.01% (w/v) $ZnSO_4 \cdot 7H_2O$ per litre, pH 7.0), Middlebrook 7H9 liquid medium (HiMedia, India) and RPMI (Thermo Fisher Scientific, USA) (1:1:1) supplemented with 0.5% v/v glycerol, 0.05% v/v Tween 80 and 10% ADC (HiMedia, India) [18]. Appropriate serial dilutions of the cells (100 μL) were added to each well. Plates were sealed and incubated at 37˚C statically for 30 days. The MPN values, based on the well pattern originating from the visible bacterial growth at the corresponding dilution point, were calculated [19].

## Measurement of L-alanine and L-glycine dehydrogenase activities

For the reductive amination of pyruvate (L-alanine dehydrogenase activity), the photometric measurement of the rate of NADH decrease was carried out. The reaction mixture consisted of 20 mM pyruvate and 0.5 mM NADH in 50 mM $NH_4OH/(NH_4)_2SO_4$ (pH 7.6). L-glycine dehydrogenase activity was measured similarly, however, except for pyruvate, 20 mM glyoxylate was used. The activity was defined as turnover of 1 μM of NADH/NAD+ in per *s*, per mg of whole cell extract.

## Preparation of samples for proteome analysis

Active and dormant cells were prepared in three biological replicates. Bacteria were harvested by centrifugation at 8000 *g* for 15 min and washed 10 times with a buffer containing (per litre) 8 g NaCl, 0.2 g KCl and 0.24 g $Na_2HPO_4$ (pH 7.4). The bacterial pellet was re-suspended in ice-cold 100 mM HEPES (4-(2-hydroxyethyl)-1-piperazineethanesulphonic acid) buffer (pH 8.0) containing complete protease inhibitor cocktail (Sigma, USA) and PMSF (phenylmethanesulphonyl fluoride) then disrupted with zirconium beads on a bead beater homogenizer (MP Biomedicals FastPrep-24) for 1 min, 5 times for active cells and 10 times for dormant cells. The bacterial lysate was centrifuged at 25,000 *g* for 15 min at 4˚C. To maximize protein isolation, SDS (2% w/v) extraction was carried out. The extracts were precipitated using a ReadyPrep 2D Cleanup kit (Bio-Rad, USA) to remove ionic contaminants such as detergents, lipids and phenolic compounds from protein samples.

## In-solution digestion

The protein extracts were dissolved in 20 μL of 6 M urea in 0.1 M ammonium bicarbonate. The protein concentration was measured using a BCA Protein Assay Kit (Thermo Fisher Scientific, Waltham, MA, USA). An equal amount of the total proteins was used for further trypsinolysis and profiling.

Proteins were reduced using TCEP (Tris-(2-carboxyethyl)-phosphin) at 25˚C for 45 min, following alkylation with iodoacetamide (IAA) in the dark for 30 min, having final concentrations of 5 mM and 55 mM, respectively. Excess IAA was quenched with 1,4-dithiothreitol, then 50 mM ammonium bicarbonate was added to a total volume of 200 μL. Trypsin was added in a protein-to-trypsin ratio of 50:1, and tryptic digestion was done at 37˚C overnight. The reaction was stopped by adding formic acid (FA) to a final concentration of 5%. The

peptide mixtures obtained were pre-fractionated and purified using a C18 Empore™ disk (3M, St. Paul, USA) as described in [20].

## LC-MS/MS analyses

Prior to analysis, peptides were dissolved in 20 μL of 3% ACN/0.1% FA. The injection volume was 1 μL with a flow rate of 5 μL/min. For the mobile phases, 0.1% FA in MS-safe water (A) and 0.1% FA in 100% ACN (B) were used.

Analysis of peptides was done on a Synapt G2-Si High Definition Mass Spectrometer employing T-wave ion mobility powered mass spectrometry coupled to an ACQUITY UPLC M-Class System (Waters) using data-independent acquisition.

The UPLC system was equipped with a trapping column (nanoEase MZ Symmetry C18 Trap Column, 180 μm × 20 mm, 5.0 μm particle diameter, 100 Å pore size, Waters) and an analytical column (nanoEase MZ HSS T3 Column, 75 μm × 100 mm, C18, 1.8 μm particle diameter, 100 Å pore size, Waters). The sample was loaded onto the trapping column for 2 min and transferred to the analytical column. The peptides were eluted from the analytical column with a flow rate of 0.4 μL/min with a linear gradient of increasing concentration of mobile phase B from 5% to 35% for 70 min. Within the nanoFlow™ ESI-source, a capillary voltage of 2.5 kV was used, the sampling cone voltage was 40 V and the source offset was 80 V. The source temperature was set to 80°C. In the trapping, ion mobility and transfer chambers, the wave velocity of the TriWave was set to 1000, 650 and 175 m/s, respectively, and ramping wave heights of 40, 40 and 4.0 V were used, respectively. In the HDMS$^E$ mode, the collision energy was ramped to 22–45 eV. Data acquisition was carried out using MassLynx software (Waters). HDMS$^E$ mass data were acquired in low and high energy modes. Raw data were noise-reduced using the Noise Compression Tool (Waters).

The acquired data were submitted for processing and database searching using Progenesis software against the database prepared in-house containing protein sequences from *Mycobacterium tuberculosis* strain ATCC 25618/H37Rv downloaded from the UniProt database (version 20190304) supplemented with sequences of common contaminants (Max Planck Institute of Biochemistry, Martinsried, Germany). The low-energy and high-energy threshold values were set to 150 and 50 counts, respectively. The parameters used for the database search were: enzyme specificity: trypsin, allowed missed cleavages: 2, fixed modification: carbamidomethylation (Cys), variable modifications: N-terminal protein acetylation, and oxidation (Met); the false discovery rate was 1%; minimum fragment ion matches per peptide, minimum fragment ion matches per protein and minimum peptides per protein were set to 1, 4 and 2, respectively. Proteins were considered as significant if they were identified in at least two independent biological replicates.

## Statistical analysis

Data analysis and visualization were carried out in R. The generated scripts are available from the corresponding author on reasonable request.

## Data normalization

Before the analysis, the total protein concentration was quantified and an equal amount of protein extracts was used for further trypsinolysis and profiling–pre-instrumental normalization on total protein concentration was carried out [21]. To make the profiles comparable for the subsequent statistical analysis, post-instrumental pre-treatment methods were carried out: to deal with heteroscedasticity and to make skewed distributions symmetric, $log_2$ transformation was performed. To force the samples to have the same distributions for the protein intensities, quantile normalization was carried out [22].

## Principal component analysis (PCA) and hierarchical clustering

For illustration of intragroup and between-group variances, principal component analysis (PCA) was carried out, applying the preliminarily normalized, $log_2$-transformed and Pareto-scaled datasets [23–25]. Similarity/dissimilarity between the groups was determined by a method of hierarchical clustering using the method of Ward [26].

## Z-score calculations and heatmap visualization

A pre-treated protein matrix, consisting of proteins arranged in rows and groups of samples arranged into columns, was centred and scaled [27]. The centring represents the difference between the value of a protein's abundancy and the total mean of the protein. The scaling is the result of division of the scaled metabolite by its σ. The resulting Z-scores, being in a similar range for the palette of proteins, make feasible comparison of the variances between the variables (proteins).

## Statistical significance analysis

The corresponding *p*-values of comparison of the two groups of cells (dormant vs active) were calculated from a two-sided *t*-test. Since the data matrix was initially $log_2$-transformed, the resulting $log_2(FC_{dorm/mean})$ is the difference between the $log_2(mean_{dorm})$ and $log_2(mean_{act})$.

## Subtractive proteomic analysis of mycobacterial proteins prevailing in dormancy with antigenic properties

The basic principles of subtractive analysis developed previously were used in the current work [28,29]. The significantly changed proteins revealing in dormancy a positive fold change (FC) were subjected to subtractive analysis. In the first step, the paralogue proteins were identified using a CD-HIT online platform [30,31]. The proteins with 60% identity were considered to be paralogous and discarded from the further analytical steps. In the next step, the BLASTp procedure was applied to select non-homologues to *Homo sapiens* proteins in the query set, using a threshold expectation value of 0.001 and similarity below 35%. The resulting list of proteins was further categorized according to their subcellular localization using the CELLO online platform [32].

The shortened list of proteins was further compared with previously published datasets with annotated *Mtb* proteins with antigenic properties: i) a list of 181 proteins detected exclusively in the sera of TB-positive patients from a study on the dynamic antibody response to the proteome [33]; ii) a list of 62 proteins resulting from serodiagnosis of latent *Mtb* infection [34]; iii) a list of 22 proteins resulting from the analysis of TB-specific IgG antibody profiles, detectable on two different platforms (Luminex and MBio) [35].

## Annotation of the list of enzymes–prodrug activators

KEGG API was used to prescribe EC numbers to the initially prescribed UniProt KO code [36].

## Results

### Dormant mycobacterial cells subjected to the proteomic analysis and the results of proteomic profiling

Dormant *Mtb* cells subjected to the further proteomic profiling bore the typical traits of 'non-culturability' [13,18], particularly incapability to grow on rich solid medium: the

experimentally estimated viability by the CFU method was $1.32 \pm 0.48 \times 10^4$ cells/mL ($\pm$ SE) for dormant cells and $1.17 \pm 0.44 \times 10^8$ cells/mL ($\pm$ SE) for active cells. Analysis of the 16S RNA sequences of cultures revealed 100% identity to *M. tuberculosis*. At the same time, estimation of the total number of viable but 'non-culturable' cells after resuscitation in fresh liquid medium by the MPN assay resulted in $1.21 \pm 0.65 \times 10^9$ cells/mL ($\pm$ SE) for dormant cells and $0.97 \pm 0.44 \times 10^9$ cells/mL ($\pm$ SE) for actively growing cells (Fig 1). Before the analysis, the total protein concentration was quantified as $3.56 \pm 0.24$ mg/g wet weight ($\pm$ SE) for dormant cells and $6.01 \pm 0.14$ mg/g wet weight ($\pm$ SE) for active cells. An equal amount of the total proteins was used for further trypsinolysis and LC-MS/MS profiling [21]. LC-MS profiling resulted in the detection of 1379 proteins (whose encoding genes Rv were further used as identifiers), covering 35% of the coding *M. tuberculosis* H37Rv genome.

However, the raw instrumental data showed skewed distributions of protein intensity, tending to lower values (Fig 2A).

For the subsequent statistical analysis, post-instrumental pre-treatment methods were carried out: to deal with heteroscedasticity and to make skewed distributions symmetric, $log_2$-transformation was performed. To force the samples to have the same distributions for the protein intensities, quantile normalization was carried out [22], resulting in comparable normally distributed profiles with equal medians, making the protein profiles eligible for further statistical analyses (Fig 2B).

## PCA and hierarchical clustering

The results of PCA [25] depict ca. 80% of all variance in the protein profiles. To perform PCA, the data matrix was preliminary Pareto-scaled [24]. The results demonstrate that samples are clearly distinguished by their physiological state ('dormant' vs 'active') (Fig 3A). Hierarchical cluster analysis, based on calculations of the amount of within-cluster dissimilarity analogously results in two distinct clusters, separating dormant cells from active ones. Ward's minimum variance method was used for the calculations [26] (Fig 3B).

## Differential analysis

The results of differential analysis are listed in the (S1 Table "Results of differential analysis") and graphically depicted as a volcano plot (Fig 4A). Generally, the abundancy of 894 proteins (out of 1379) was found to change significantly ($p < 0.05$) in the transition to the dormant state–among them 468 with a positive FC ('stored in dormancy proteins') and 426 with a negative FC ('decreased in dormancy proteins'). The rest of the proteins (485) lay beyond the scope of differential analysis–there was greater variance in their fluctuation in transition to dormancy and consequently their changes were insignificant. Analysis of the first 30 proteins stored up significantly differently in dormancy, bearing the maximum $log_2FC$, discloses a cohort of regulatory proteins and virulence factors (and not enzymatic proteins), implying biochemical adaptation to the stress conditions (Fig 4B). According to the main stress factor–pH decrease–we observed upregulation of the transcriptional regulator PhoP (*Rv0757*, $p < 0.05$, $log_2FC = 1.43$) of the two-component PhoP–PhoR system, conferring the fitness of mycobacteria to the lowered pH values [18,37,38]. In addition, PhoPR controls secretion of the antigens Ag85A (*Rv3804c*, $p < 0.05$, $log_2FC = 1.28$) and Ag85B (*Rv0129c*, $p > 0.05$, $log_2FC = 1.11$) as well as ESAT-6 EsxA (*Rv3875*, $p < 0.05$, $log_2FC = 4.14$)–a secreted strong human T-cell antigen protein that plays a number of roles in modulating the host's immune response to infection [39] (Fig 4C). Tightly associated with the ESX-1 and ESX-5 secretion systems are proteins of the PE/PPE family [40]. Out of 95 KEGG-annotated PE/PPE proteins we detected six: PPE18 (*Rv1196*, $p < 0.05$, $log_2FC = 2.44$), PPE19 (*Rv1361c*, $p < 0.05$, $log_2FC = 2.11$), PPE32 (*Rv1808*,

**A**

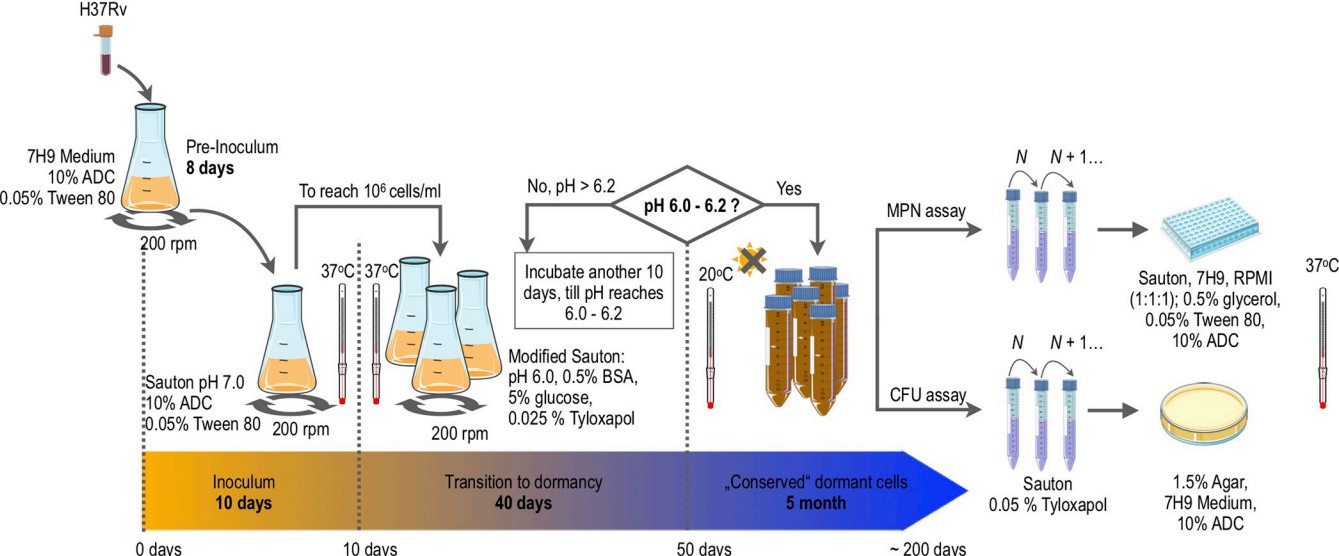

**B**

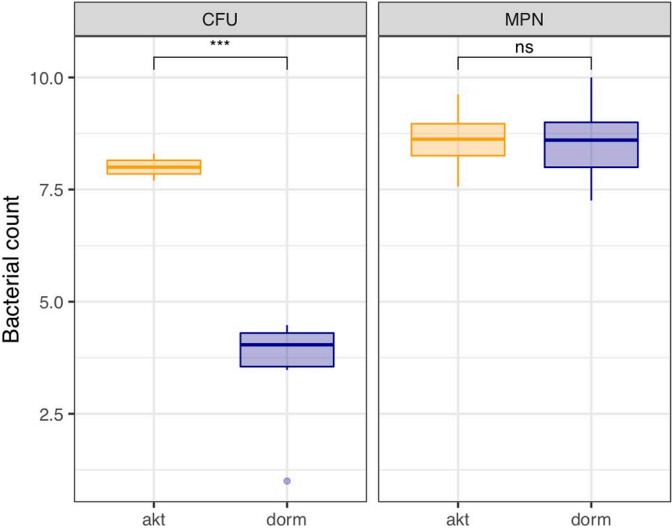

**Fig 1. General scheme of dormant cells production and viability assessment of *M. tuberculosis* H37Rv cells subjected to further proteomic analysis.** A) The outline of the model of gradual acidification of the bacterial culture in the post-stationary phase used in the current study [13]. See details in the Material and Methods section; B) The results of viability assays CFU and MPN of Active cells (akt) were probed in the late log phase–the 10th day of cultivation in unmodified Sauton medium. Viability of the dormant cells (dorm) was analysed after 5 months of storage after gradual acidification of the bacterial culture in the post-stationary phase. Estimation based on probing of 6–9 independent samples of each type of cell. Statistically significant decrease (***—*p* 0.001) in the CFU value at dormant state, while maintaining the unchanged MPN value (ns–non-significant), indicates the transition of bacteria to the state of viability, but "non-culturability". Hereinafter, unless otherwise stated, the group of dormant cells is colored blue, the group of active cells is colored orange.

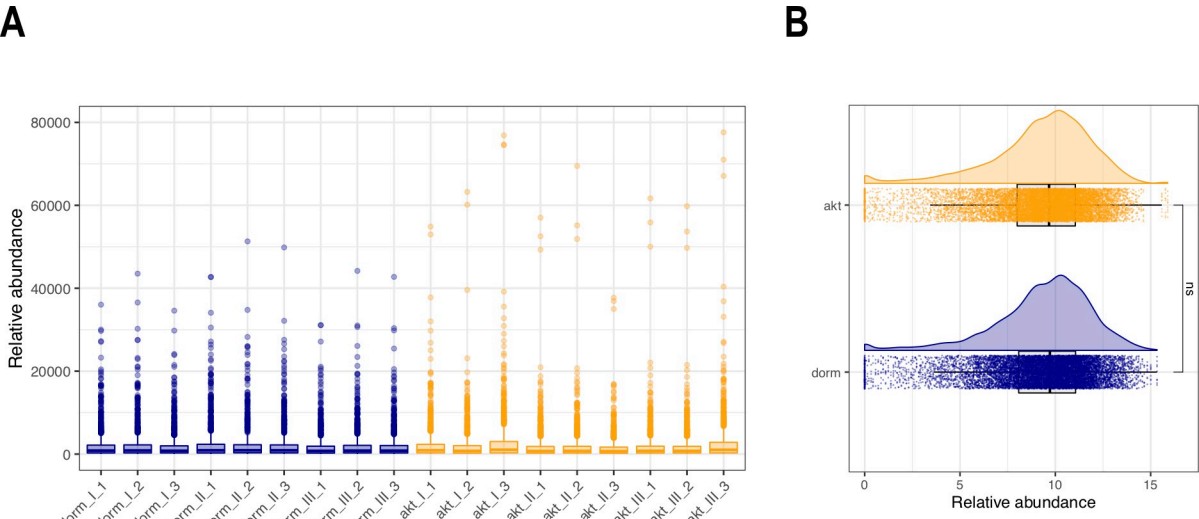

**Fig 2. Normalization of proteomic profiles of dormant (5 month of storage) and active *M. tuberculosis* cells.** A) Distribution of raw protein intensities demonstrates skewed distributions of protein intensity, tending to lower values; B) average median intensity of nine samples (biological and technical replicates) for each physiological condition after $log_2$ transformation and quantile normalization of the protein profiles results in comparable normally distributed profiles with equal medians, making the protein profiles eligible for further statistical analyses.

$p < 0.05$, $log_2FC = 3.15$), PPE60 (*Rv3478*, $p < 0.05$, $log_2FC = 1.05$), PE15 (*Rv1386*, $p < 0.05$, $log_2FC = 3.64$) and PE31 (*Rv3477*, $p > 0.05$, $log_2FC = 0.39$). The low discovery rate may be explained by the paucity of trypsin-cleavage sites in these proteins and their structural homology [41]. For the proteins of this class, a variety of functions is assumed: many PE and PPE

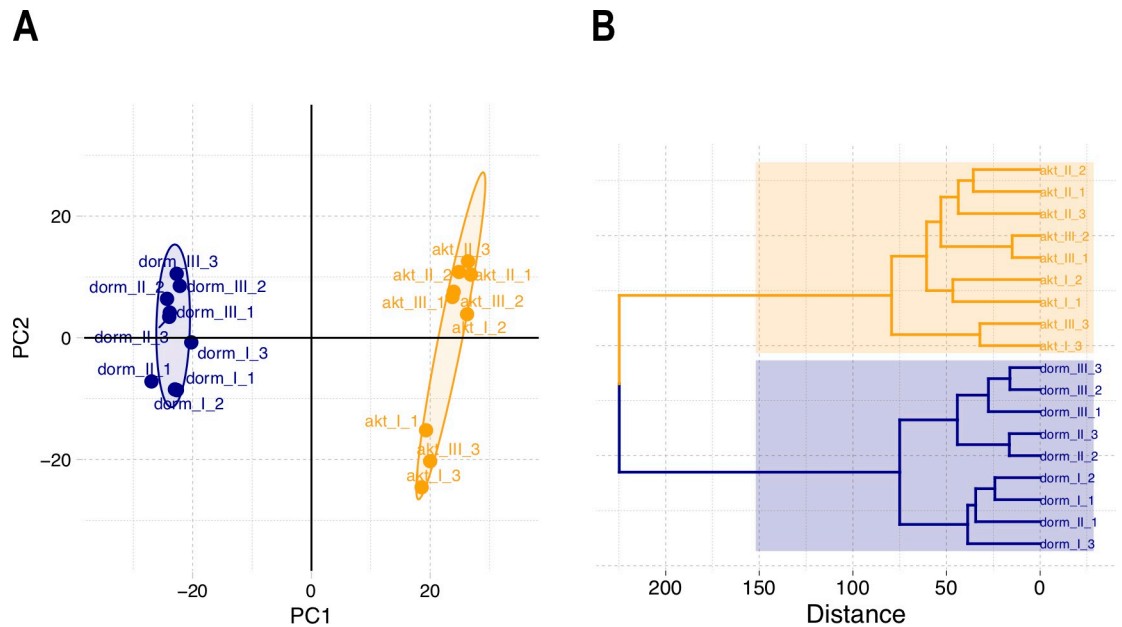

**Fig 3.** Results of the principal component analysis (A) and hierarchical clustering (B) of *M. tuberculosis* cells used for proteomic profiling. PC1 with the largest variance of each protein level separates the samples into two distinct clusters. The elliptical boundary is analogous to the 95% CI for both bivariate distributions. Ward's method [26] of hierarchical clustering similarly demonstrates segregation of the two types of physiological group.

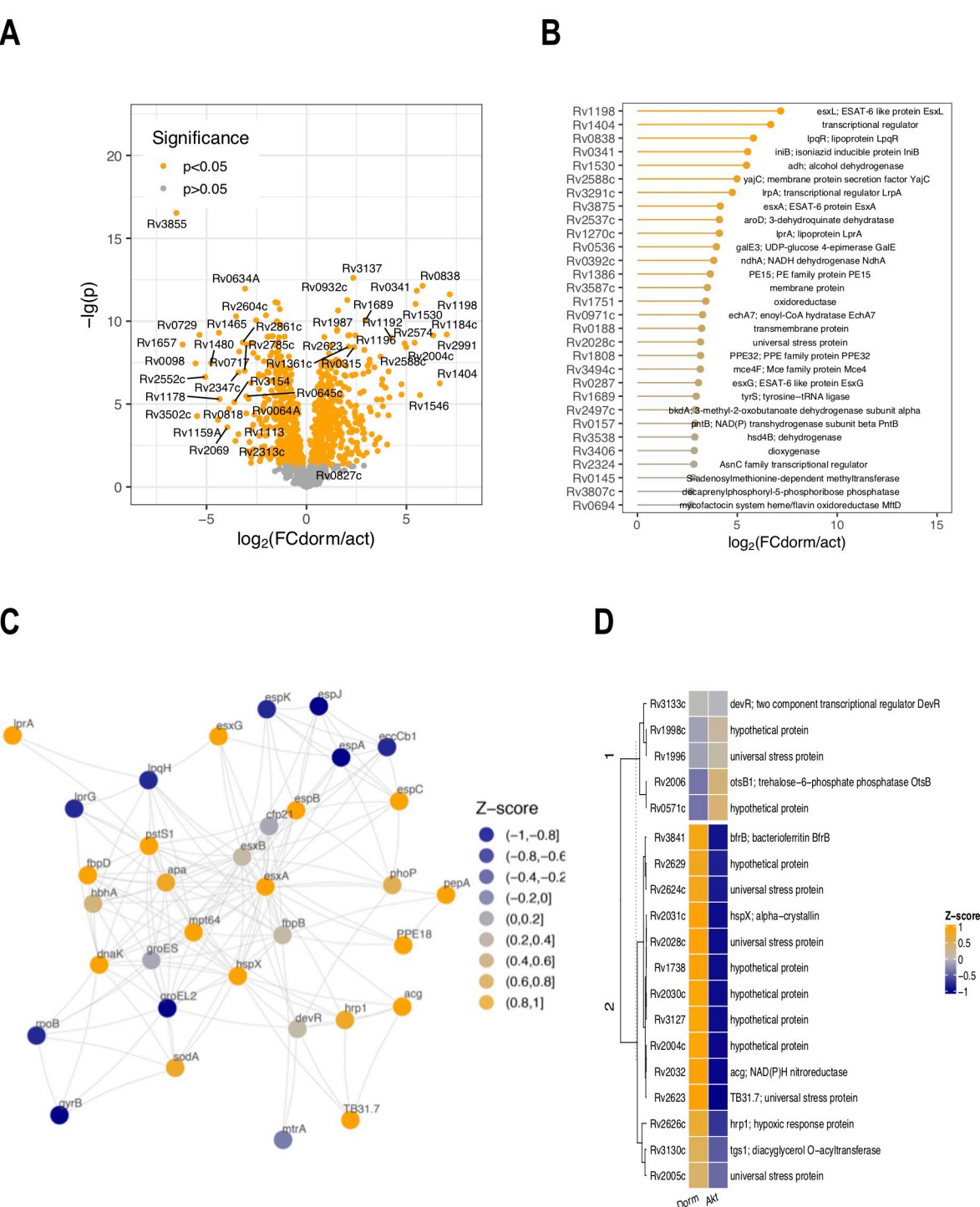

**Fig 4. General statistical results of comparative proteomic analysis of dormant vs active *M. tuberculosis* cells.** A) Results of differential comparison of dormant cells vs active cells reveal 894 statistically significantly changed proteins, among them 468 proteins prevailing in dormancy (with a positive FC value) and 426 proteins whose abundancy decreased in the transition to dormancy (with a negative FC value); B) visualization of the 30 proteins most abundant in dormancy with the maximum FC; C) STRING-based interaction network of PhoP and DevR regulators. Upregulation of pH-dependent regulator PhoP suppresses DevR, a regulator of the DosR system, however activating the expression of mycobacterial virulence factors: ESAT and PE/PPE proteins; D) heat map of the detected proteins of DosR regulon (14 upregulated out of 50 postulated for Wayne's model of dormancy) [46].

genes have been shown to be upregulated in starvation and during stress conditions [42]. A chaperone function of PE proteins has been reported, particularly in transporting of MPT64 antigen protein (*Rv1980c*, $p < 0.05$, $log_2FC = 0.69$) [43]. Additionally, the PhoP transcriptional regulator has been shown to downregulate DevR (*Rv3133c*, $p > 0.05$, $log_2FC = 0.01$), a regulator of the DosR 'dormancy' regulon [37,44]. Correspondingly, out of ca. 50 postulated genes of the DosR regulon, only 19 proteins were detected in the current study, 14 of which were found significantly stored in the dormant state (Fig 4D). Similarly, we observed a down-shift of a number of other transcriptional regulatory proteins, e.g., *Rv0275c*, *Rv1556*, *Rv1019*, *Rv0818*, *Rv3058c*, *Rv3249c*, etc. Analogously, in the absence of translational processes, the abundancy of translation initiation factors (*Rv3462c*, *Rv2839c*) as well as 30s and 50s ribosomal proteins (*Rv0717*, *Rv2875c*, *Rv0641*) was similarly decreased [45].

There was no consistency in behaviour of proteins of the ABC trehalose transporter complex LpqY–SugA–SugB–SugC (*Rv1235*, *Rv1236*, *Rv1237*, *Rv1238*): out of four annotated in the *Mtb* genome only *Rv1235* and *Rv1238* were detected in the current study, albeit with a different sign of FC in dormancy, rather indicating a decrease in their abundancy and efficacy in the current stress conditions and confirming the dependency of the dormant cells on trehalose to replenish intracellular pathways–particularly glycolysis and pentose phosphate pathway [47,48]. On the contrary, the components of the phosphate ABC transporter PstS1-3 (*Rv0928*, *Rv0934*, *Rv0932c*) were found concordantly to be stored in dormancy, indicating a possible dependency of dormant cells on extracellular sources of inorganic phosphate. Part of the ABC efflux pump *Rv1218c*, conferring the cells the resistance to a wide range of drugs, was found to be similarly stored.

For the analysis of biochemical adaptation to the transition to dormancy, the 666 detected enzymes (both with positive and negative $log_2FC$, based on KEGG orthology annotation) were placed on the metabolic map and the corresponding processes which they catalyse were highlighted: in orange for those enzymes with a positive FC (enzymes stored in dormancy) and in blue for those demonstrating a negative FC (a relative decrease in dormancy). The width of the highlighted lines corresponds to the *p*-value (the thicker lines are for $p < 0.01$, the thinnest lines are for insignificant results ($p > 0.05$) (Fig 5A). A glance at the general metabolic visualization (Fig 5A) gives an impression that, in dormancy, a scattered 'mosaic' intactness of enzymes can be seen: the flawless functionality of the many pathways is hindered by enzymes significantly decreased in dormancy, e.g., the functionality of the urea cycle may be broken because of the decreased level of argininosuccinate synthase (*Rv1658*, $p < 0.05$, $log_2FC = -0.69$). To annotate precisely the metabolic pathways differentially preserved in dormancy, pathway enrichment analysis was carried out on the enzymatic proteins with FC > 0 and $p < 0.05$ [49,50] (Fig 5B). Detailed pathway analysis revealed nine upregulated proteins of TCA; however, isocitrate dehydrogenase (*Rv0066c*, $p < 0.05$, $log_2FC = -0.41$) and subunits of α-ketoglutarate dehydrogenase (*Rv2454* and *Rv2455*) demonstrated negative values of $log_2FC$, impeding the normal flow of the TCA cycle (Fig 5C). The subunits of membrane-bound fumarate reductase were not technically detected; however, subunits of the succinate dehydrogenase complex (*Rv3318* and *Rv0247c*) were detected, potentially enabling the reverse flow of the processes in the 'reductive branch' of the TCA cycle (part of the Arnon–Buchanan cycle) from pyruvate to succinate, providing the cell with reoxidized $NAD^+$ [51,52]. To the NADH regeneration may contribute L-alanine dehydrogenase (catalysing the reductive amination of pyruvate to L-alanine) and glycine dehydrogenase (catalysing the reductive amination of glyoxylate to glycine), whose activity was shown to increase in microaerophilic conditions[53,54] and was confirmed experimentally in the currents study: the activity of the former was 27.87±5.58 $\mu M(NADH) \cdot s^{-1} \cdot mg_{extract}^{-1}$ (± SE) for active cells and 63.82 ±1.28 $\mu M(NADH) \cdot s^{-1} \cdot mg_{extract}^{-1}$ (± SE) for dormant cells, and the activity of the latter was 15.59 ±2.83 $\mu M(NADH) \cdot s^{-1} \cdot mg_{extract}^{-1}$ (± SE) for active cells and 15.08±1.46 $\mu M(NADH) \cdot s^{-1} \cdot mg_{extract}^{-1}$ (± SE) for dormant cells.

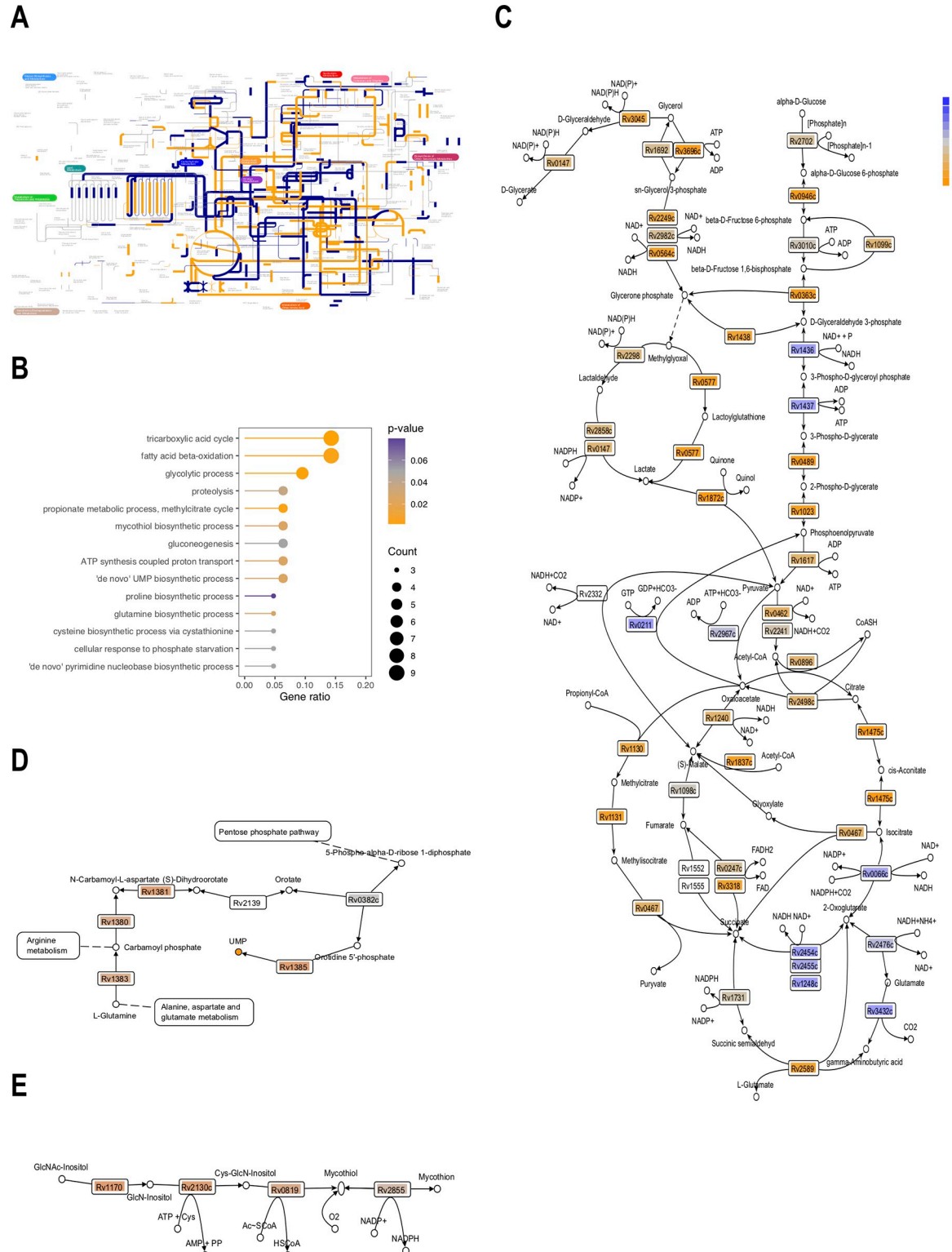

**Fig 5. Proteomic composition of enzymatic proteins in dormant *M. tuberculosis*.** A) Pathway visualization, based on KO annotation: the detected enzymes (either stored in the dormant state, or those whose abundancy decreased) were placed on a metabolic map and the corresponding processes highlighted–orange for enzymes stored in dormancy (FC > 0), dark blue for degraded enzymes (FC < 0); line thickness corresponds to statistical significance–the thickest lines depict statistically valuable processes; B) GO enrichment analysis for the pathways assembled by the enzymes stored in dormancy. Gene ratio–the ratio of the annotated involved genes of the concrete pathway to

the total annotated genes of these pathways; C) reconstruction of central carbon metabolism pathways of dormant *M. tuberculosis* cells–Z-scores are mapped on the corresponding Rv coding enzymes; D) reconstruction of '*de novo*' UMP pathway. UMP accumulation in the dormant state was similarly recently observed in the same model conditions in *M. smegmatis* cells [55]; E) mycothiol salvage pathway.

The intactness of enzymes acting in the β-oxidation of fatty acids may provide the cell with a surplus of acetyl-CoA (in the case of oxidation of even-chained fatty acids) and propionyl-CoA (in the case of oxidation of odd-chained fatty acids). Dormant *Mtb* cells demonstrated the presence of enzymes of the methylcitrate cycle, which may allow the conversion of propionyl-CoA and oxaloacetate to succinate and pyruvate. The latter can be converted to fumarate by the detected succinate dehydrogenase (*Rv3318*, *Rv0247c*). If this pathway is functioning, the accumulation of organic acids (contributing to sustainment of the energized level of transmembrane potential in dormancy [56,57]) in the cells and medium should be expected; that, however, was found experimentally for dormant cells of *M. smegmatis*, obtained in the same model of self-acidification of growth medium [55]. Contributing to the accumulation of these acids and to detoxification of surplus acetyl-CoA is the glyoxylate pathway, whose enzymes isocitrate lyase (*Rv0467*) and malate synthase (*Rv1837c*) were found to be accumulated in dormant cells.

The classical flow of the glycolytic pathway is hampered by deficient glyceraldehyde-3-phosphate dehydrogenase (*Rv1436*, $p < 0.05$, $log_2FC = -0.43$) and phosphoglycerate kinase (*Rv1437*, $p < 0.05$, $log_2FC = -0.61$)–what is supported by metabolomic studies [58]; however, there is a bypass, consisting of the formation of glycerone phosphate, which can further spontaneously convert into methylglyoxal, which in further detoxification pathways leads to pyruvate through lactate formation. Similarly, all enzymes of glycerol metabolism were detected–the catabolism of glycerol or glycerol-3-phosphate is conjugated with the toxic carbonyl compounds methylglyoxal and lactaldehyde (the latter was detected experimentally for *M. smegmatis* [55]). However, functioning aldehyde dehydrogenases (*Rv0147*, *Rv2858c*) or glyoxylase (*Rv0577*) detoxify these intermediates, ultimately to lactate. The further transformation of forming lactate by a quinone-dependent lactate dehydrogenase finally supplies the cells with pyruvate capable of further catabolism in the TCA, glyoxylate and methylcitrate pathways. *Mtb* used to have annotated two lactate dehydrogenases, LldD2 (*Rv1872c*, $p < 0.05$, $log_2FC = 1.04$) and LldD1 (*Rv0694*, $p < 0.05$, $log_2FC = 2.67$); however, lactate dehydrogenase activity was confirmed for LldD2 only [59]. On the other hand, upregulation of *Rv0694* was observed in the current conditions of dormancy in glycerol-glucose medium. *Rv0694* is a member of the *Rv0691–Rv0694* gene cluster, involved in biosynthesis of a recently discovered peptide-derived alternative electron carrier–mycofactocin [60,61].

The decreased detection of particular enzymes may be due to the preservation of proteolytic peptidases abundant in the dormant proteome: *Rv0319* ($p < 0.05$, $log_2FC = 1.1$), *Rv0457c* ($p < 0.05$, $log_2FC = 0.82$), *Rv3883c* ($p < 0.05$, $log_2FC = 2.53$) and *Rv0983* ($p < 0.05$, $log_2FC = 0.64$).

The enzymes of glycerol catabolism that are stored unprocessed in dormancy contribute to lipid metabolism and may serve as a source of reoxidation of reductive equivalents (Fig 5C). The gapped glycolytic pathway fails to serve substrate-level ATP generation (if any): however, accumulated polyphosphate (the transport of inorganic phosphate remains intact) may serve a phosphagen–a reservoir of phosphoryl groups, that can be used to generate ATP, thus we detected a reversible polyphosphate kinase Ppk1 (*Rv2984*, $p < 0.05$, $log_2FC = 0.57$).

Additionally, we were able to reconstruct a *de novo* unimpaired uridine monophosphate (UMP) pathway, serving the precursor of all pyrimidine nucleotides (Fig 5D).

Similarly to the known involvement of protective redox systems as a response to ROS attacks in macrophages, upregulation of superoxide dismutase was observed in our dormancy

conditions SodA (*Rv3846*, $p < 0.05$, $log_2FC = 1.04$). Since mycobacteria lack glutathione, the ubiquitous low-molecular weight thiol in other organisms, *Mtb* is dependent on mycothiol for antioxidant activity; the enzymes encoding its biosynthesis were found to be stored in the dormant state (Fig 5E).

To the oxidative stress protective system belong $F_{420}$-dependent glucose-6-phosphate dehydrogenase Fgd1 (*Rv0407*, $p > 0.05$, $log_2FC = 0.71$) and $F_{420}H_2$-dependent menaquinone reductase *Rv1261c* ($p > 0.05$, $log_2FC = 0.52$), contributing to the protection of bacteria from the formation of toxic semiquinones [62]. In shaping the response to ROS and RNS, the accumulation of the regulator SigH was observed (*Rv3223c*, $p < 0.05$, $log_2FC = 0.48$).

## Annotation of immunogenic proteins accumulated in dormant cells

It was interesting to search for immunogenic proteins among the detected proteins with a positive FC in dormant *Mtb*, as they are potentially applicable for latent TB diagnosis. In the current study we enriched the possible candidates, based on the subtractive proteomic concept [28]. The essence of this approach presumes the identification of non-paralogue proteins followed by the identification of non-homologues to *H. sapiens* proteins and the analysis of subcellular localization (Fig 6A). Thus, the initially applied 468 proteins significantly represented in dormancy with a positive FC resulted in 273 non-homologue proteins after the first two filters, further localization analysis of which allowed the identification of 53 membrane proteins, 175 cytoplasmic proteins and 45 extracellular proteins.

The short-listed data set was compared with the reference lists from published works on studies of the sera of human TB patients [33–35]; (S1 Table "Antigenic proteins").

The comparative analysis reference lists comprised the most reactive proteins (in total 265 antigens) as described in the Material and Methods.

Comparative analysis of the proteins in these reference sets with the selected proteins in the current study with a positive FC in dormancy revealed 40 potentially immunogenic proteins specific for dormancy (Fig 6A). Remarkably, these studies (including our data set) demonstrate only one shared protein, *Rv1284* –a carbonic anhydrase ($p < 0.05$, $log_2FC = 0.38$), which was previously annotated among the latency-related antigens of *Mtb* [63] (Fig 6B).

## Annotation of proteins accumulated in dormancy–potential prodrug activators

The accumulation of a significant number of enzymatic proteins in the dormant state may imply the potency of 'non-culturable' *Mtb* cells to perform biotransformation reactions.

Analysis of the distribution of enzyme classes in dormancy revealed the preservation of a bulk bundle of oxidoreductases and hydrolases (Fig 7A). The list of 73 oxidoreductases detected in dormancy with an FC > 0 is summarized in the (S1 Table "Enzymatic proteins"). The majority of the annotated enzymes of this class are NAD- and FAD-dependent, bearing relatively low standard reduction potentials (−0.32 and −0.22 V correspondingly). There are, however, two $F_{420}$-dependent enzymes, *Rv0407* ($F_{420}$-dependent glucose-6-phosphate dehydrogenase Fgd1) and deazaflavin ($F_{420}$)-dependent nitroreductase (Ddn), to detect. $F_{420}$-dependent enzymes are known to bear a lower reduction potential (−0.32 V) and have been shown to participate in bicyclic nitroimidazole compounds reduction [62,64]. Similarly, the FAD-dependent decaprenylphosphoryl-beta-D-ribose-2'-epimerase (catalysing the conversion of decaprenyl-phosphoryl-D-ribose to decaprenyl-phosphoryl-D-arabinose–the sole arabinose precursor in the pathway of arabinogalactan biosynthesis) is capable of nitroreducing recently developed benzothiazinones (BTZ) [65].

**A**

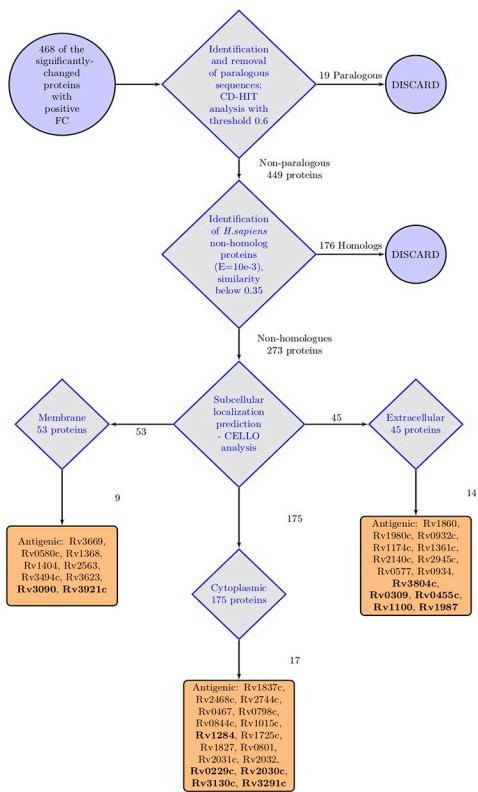

**B**

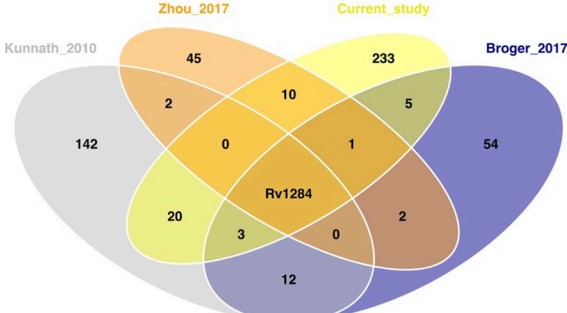

**Fig 6. Determination of potential antigenic proteins in dormant *M. tuberculosis* proteome.** A) General workflow for establishment of potential antigenic proteins. The complete list of antigenic proteins is available in the (S1 Table "Antigenic proteins"); bold-annotated proteins are those with immunogenic properties selected by the sera of latently infected patients [34]; B) Venn analysis of the reference antigenic proteins annotated from previously published works [33–35], with the list of proteins resulting from the subtractive proteomic analysis of the current data set.

The component of the non-proton pumping NADH:quinone oxidoreductase (*Rv0392c*, $p < 0.05$, $log_2FC$ = 3.81) accumulated in dormancy is hypothesized to reduce clofazimine, an antibiotic whose mode of action is mediated through ROS formation [67].

**A**

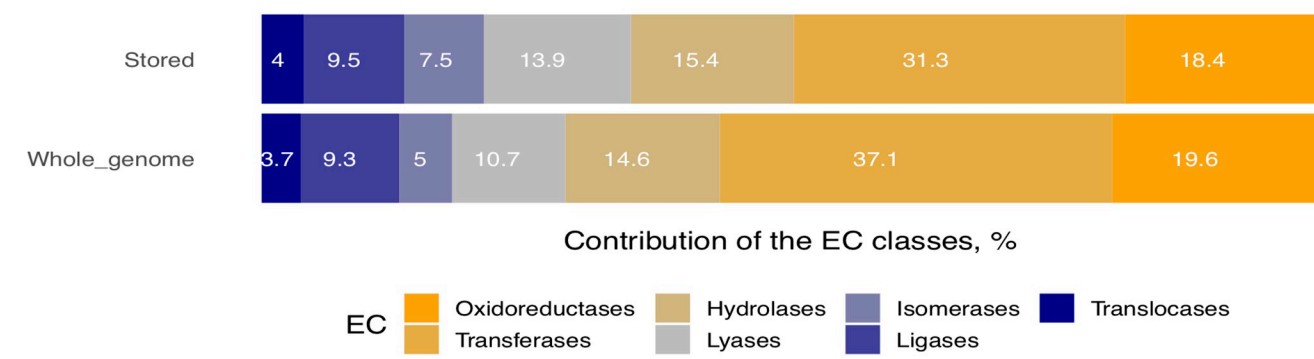

**B**

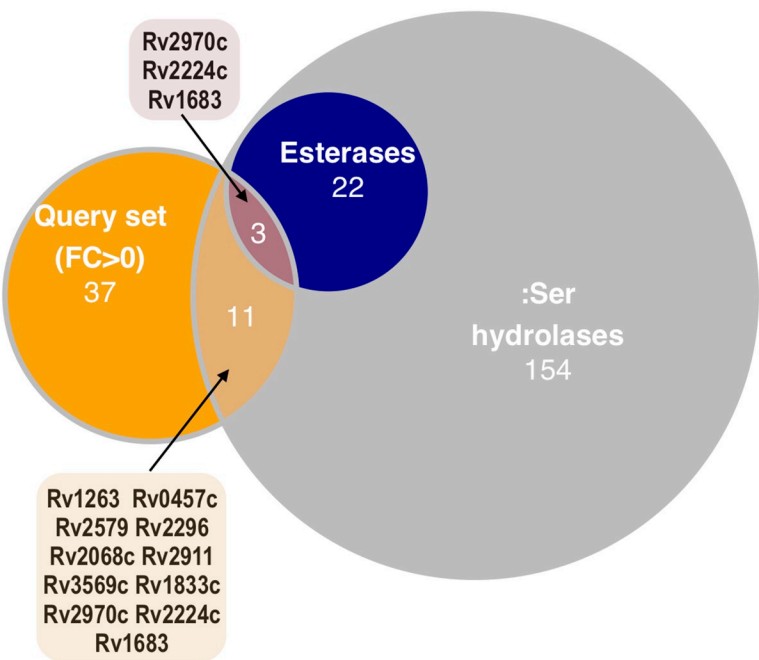

**Fig 7. Enzymes of dormancy as potential intracellular prodrugs activators.** A) Distribution of the enzyme classes within the enzymes stored in dormancy compared to whole-genome enzyme classes; B) overlap of the detected 48 hydrolytic enzymes with the annotated group of serine hydrolases and esterases [66].

Hydrolytic enzymes (esterases and lipases) are considered to play a crucial role in mycobacterial persistence [68,69]. Among the enzymes detected with a positive FC in the dormant state, 48 could be classified as hydrolytic enzymes (S1 Table "Enzymatic proteins"). This list was compared with a list of annotated serine hydrolases and esterases [66] (Fig 7B). In the query data set of enzymes with a positive FC, three enzymes potentially keep their activity in dormancy (*Rv2970c*, *Rv2224c* and *Rv1683*) [66]. The list of enzymes of other EC classes can be found in the (Table "Enzymatic proteins").

## Discussion

The observation that the majority of proteins in dormancy are significantly abundant (in comparison to the active state), and bear a positive FC, corresponds to the previously reported observation on the ability of *Mtb* to preserve protein diversity in the dormant state [18]. Comparison of the data originating from 2D electrophoresis analysis of D1 cells (4.5 months of dormancy) [18] with the same gene set from the current study (262 genes) after a common ranking procedure, followed by the Mann–Whitney test, reveals similarity between the data sets ($H_o$: $U_{LC-MS} = U_{2D}$, $p$-value = 0.8894) (Fig 8A). Similar to the previously published 2D proteome analysis, only a limited number of accumulated proteins belong to the annotated DOS regulon, which demonstrates the differences between the anoxic Wayne's model and the current model of self-acidification in the post-stationary phase. Analogously, the increased abundancy of *Rv0757* –a component of the pH-dependent regulator PhoP which is upregulated in D1 (4 month of dormancy) and D2 (1 year of dormancy) dormancy states–may contribute to the low discovery rate of proteins of the DOS system [18].

PhoP is a virulence regulator which is conserved in a wide range of bacteria and controls the transcription of more than 600 genes [70]. Besides its function of regulating the transcription of virulence factors (see the Section "Differential analysis"), it has been hypothesized that PhoP coordinates gene expression with changes in membrane potential, and encodes the proteins that control the state of membrane energization [70]. In *Salmonella*, PhoP has been shown to control the production of superoxide dismutase, SOD [71], which was similarly found to be abundant in dormancy in the current study. Correspondingly, the protective systems were detected in the current study–accumulation of SODs (SodA–*Rv3846*, SodC–*Rv0432*) and the DNA-binding proteins HupB (*Rv2986*) and IniB (*Rv0341*), as well as several chaperone proteins (DnaK–*Rv0350*, DnaJ1 –*Rv0352*). However, 2D analysis, being qualitative in its nature, could not disclose the changes in quantitative abundance of these proteins. Thus, e.g., catalase G—KatG, (*Rv1908c*) could be detected in D2 [18], the abundancy of which in the dormant state in the current study, however, was slightly decreased in comparison to the active state.

Moreover, the main drawback of the 2D approach is its inability to reach a meaningful coverage of biochemical processes–thus, GO enrichment analysis based on the enzymes detected in D1 [18] revealed only three statistically feasible pathways. Manual reconstruction of the glycolysis and TCA pathways demonstrated only a limited coverage of these pathways. Moreover, those remaining undisclosed were the transformation of glycerol, propionyl-CoA metabolism and the glyoxylate pathway (Fig 8).

The high-throughput LC-MS-based reconstruction of central carbon metabolism significantly increased the coverage of metabolic processes, and additionally disclosed mycothiol and a *de novo* UMP biosynthetic pathway (Fig 5). The annotated pathways resemble the adjustment of mycobacterial metabolic machinery for life inside lipid-rich macrophages, particularly involvement of the methylcitrate and glyoxylate pathways to utilize the excess acetyl- and propionyl-CoA (resulting from the catabolic oxidation of fatty acids, cholesterol and branched

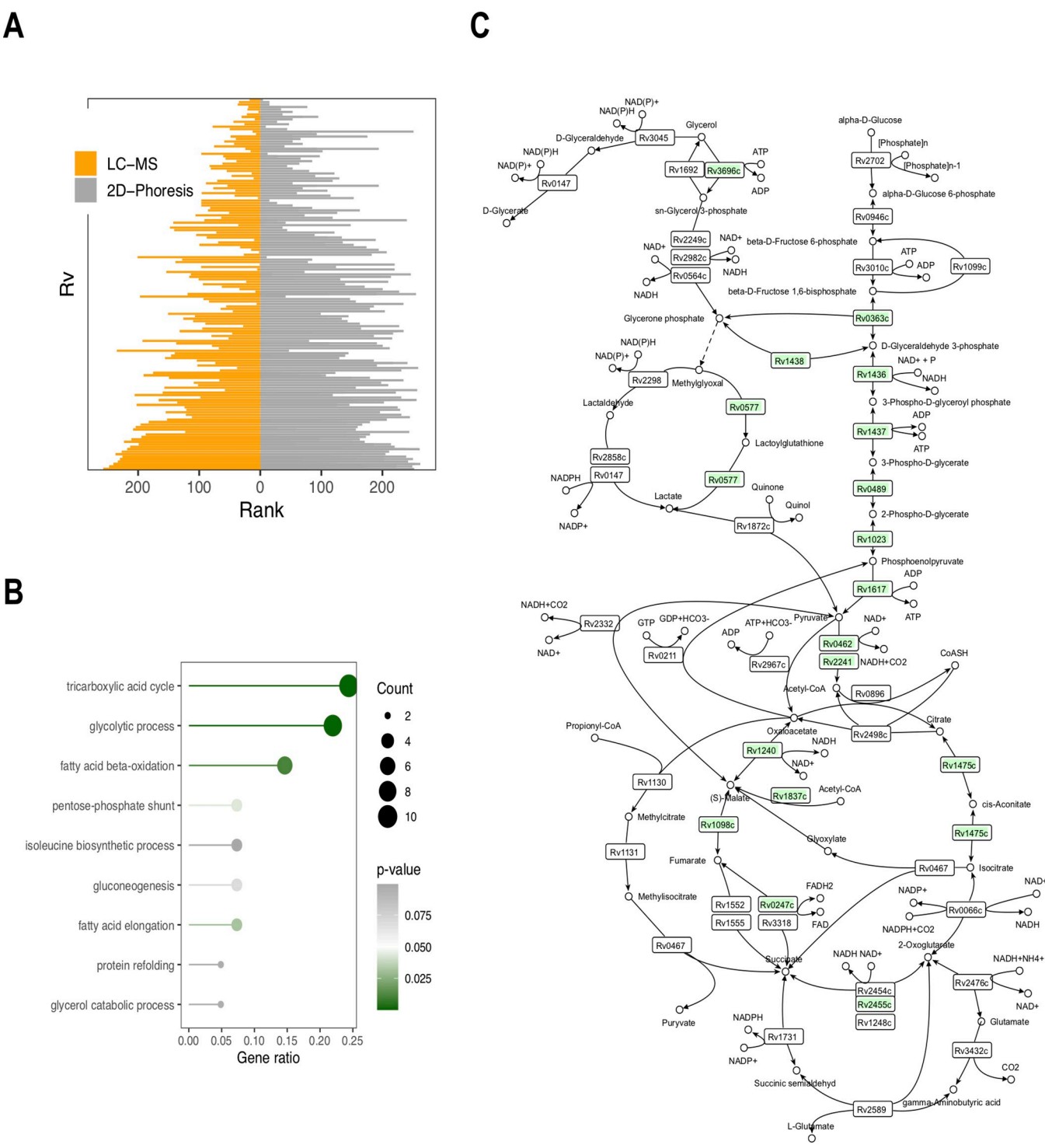

**Fig 8. Comparison of the current LC-MS data with the previously published work on 2D proteomics of dormant *M. tuberculosis* [18].** A) Visualization of the results of Mann–Whitney comparison of data acquired in the current study with the previously published data on 2D-proteomic analysis; B) GO enrichment analysis for the pathways assembled by the enzymes stored in D1 dormant state (4 month of dormancy). Gene ratio—ratio of the annotated involved genes of the concrete pathway to the total annotated genes of these pathways; C) coverage of central carbon metabolism in D1 dormant state.

amino-acid catabolism) [72–75]. It is worth noting that such biochemical adaptation is tightly linked to mycobacterial virulence; thus, isocitrate lyase, besides its biochemical functions of participation in the glyoxylate cycle, in the methyl citrate cycle of propionyl-CoA metabolism or in succinate-associated generation for maintenance of proton-motive force in dormancy [52,55], is an important mycobacterial virulence factor [72,76]. The decrease in virulence in *icl* mutants is precisely related to the enzymatic function of Icl and reflects the importance of the glyoxylate shunt for bacterial life *in vivo*.

The prevalence of catabolic enzymes in the absence of respiration may result in the generation of an excess of reduced $FADH_2$ and NADH, which should be reoxidized for the continuation of metabolism [77]. In the absence of respiration, a proposed mechanism for $NAD^+$ replenishment and maintaining the energized level of transmembrane potential consists of exploiting the reductive branch of the TCA cycle (a part of the Arnon–Buchanan cycle) [52,57], which is corroborated by the recently published data on succinate accumulation in the same model by dormant *M. smegmatis* cells [55].

Nonetheless, the current data on relative protein changes may provide only speculative information on real flux flows inside the cells [78], albeit speculatively envisaging metabolic adaptation of the cells to persistence alongside the activation of virulence factors, favouring the host's immune escape. At the same time, enzymes stored in dormancy that are involved in particular metabolic pathways may be exploited by dormant cells during reactivation, when macromolecular biosynthetic processes are initiated [79]. For example, the salvage of enzymes involved in the metabolism of purine and pyrimidine, the key precursors of DNA and RNA, in the current study (similarly to the observation of accumulation of the corresponding metabolites in the metabolism of dormant *M. smegmatis* [55]) makes these precursors immediately accessible after the onset of reactivation.

At the same time, individual enzymes preserved in dormancy can retain functionality, in particular those with cytoprotective functions.

Thus, under excess exogenous glycerol one would expect the accumulation of carbonyl compounds, e.g., lactaldehyde [55] or methylglyoxal (Fig 5). Therefore, it is to be expected that mycobacteria have developed mechanisms of protection from such reactive electrophilic agents.

Functioning NAD(P)H-dependent aldehyde dehydrogenases ensure the flow of metabolic transformations of carbonyl compounds and, on the other hand, may represent valuable targets for TB drug discovery [80]. The glyoxylase *Rv0577*, presumably ensuring the detoxification of methylglyoxal, has similarly been considered as a target for pyrimidine imidazoles [81].

Whereas a majority of organisms (including human beings) exploit glutathione as a carbonyl detoxification system, mycobacteria lack glutathione, being dependent on mycothiol, a conjugate of N-acetylcysteine with *myo*-inosityl-2-amino-deoxy-α-D-glucopyranoside [82]. Aside from its antioxidant properties, mycothiol may act as a cofactor for aldehyde dehydrogenase and participate in detoxification of rifampicin and isoniazid [82,83]. Correspondingly, pathway enrichment analysis disclosed the salvage of biosynthetic enzymes of mycothiol biosynthesis (Fig 5E).

Part of the adaptation to oxidative stress is the accumulation of $F_{420}$-dependent enzymes: glucose-6-phosphate dehydrogenase–Fgd1 (*Rv0407*) and $F_{420}H_2$-dependent quinone reductase (deazaflavin-dependent nitroreductase, *Rv1261c*). The first passes electrons from the resulted in gluconeogenesis glucose-6-phosphate on $F_{420}H_2$, which further participates in specific two-electron reduction of quinones by the $F_{420}H_2$-dependent quinone reductase (deazaflavin-dependent nitroreductase, *Rv1261c*), preventing the one-electron reduction pathway and the formation of cytotoxic semiquinones, notorious for formation of superoxide radical. The evolution of this self-protective mechanism was necessary, considering menaquinone as a main

quinone of mycobacteria, which due to its more negative redox potential than ubiquinones tends to generate superoxide [84].

Thus, alongside the individually functioning SODs, catalases and chaperones, mycobacterial cells are armoured with efficient defence systems, protecting the bacteria against various stress factors.

Therefore, the current model of self-acidification of *Mtb* in the post-stationary phase being an *in vitro* model, it mirrors the virulence and biochemical adaptation of these bacteria to *in vivo* persistence.

Comparative analysis of the proteins stored in dormancy found in the present study with the recently established immunogenic *Mtb* proteins resulted in a list of 40 proteins with immunogenic properties (Fig 6).

Despite the serodiagnosis of latent TB being very obscure, the work of Zhou et al. [34], where serodiagnostic analysis was carried out using the sera of patients with confirmed latent TB at the proteome level, is of particular interest. Among 62 seropositive TB proteins reported by Zhou et al., 12 proteins out of 40 were found in the current study. Additionally, experiments recently carried out on the sera of 42 TB patients allowed the detection of a set of 27 potentially immunoreactive proteins [85], six of them (*Rv2623*, *Rv2018*, *Rv2145c*, *Rv2744c*, *Rv1837c*, *Rv0341*) similarly detected in the current study with a positive FC in the dormant state. It is to be anticipated that the list of selected antigenic proteins includes those that may be of interest for identifying the specific 'immunosignature' of latent TB, which can later be used for immunodiagnosis of latent TB. However, this assumption requires further experimental verification.

The currently available therapy with the commonly exploited 'first- and second-line' TB drugs is directed at a rather narrow spectrum of *Mtb* targets, controlling the processes of DNA replication and transcription translation or targeting the enzymes of cell-wall biosynthesis/remodelling. However, all these 'targets' require the active flow of physiological pathways in the bacterial cells, sparing therefore the population of metabolically inactive dormant cells. Therefore, accounting for the urgent need for new chemicals efficient against drug-resistant mycobacteria and drug-tolerant dormant populations, it is an attractive idea to consider the enzymes available in dormancy capable of biotransformation, which could activate prodrugs and result in the accumulation of substances with general toxicity (ROS, RNS and chemically reactive metabolites and antimetabolites) [11,12].

Some prodrugs such as clofazimine or delamanid which are activated by intracellular specific cofactor-dependent enzymes may serve as an illustration of this approach. The first molecule generates intracellular ROS through NADH dehydrogenase [67]. The latter undergoes a metabolic conversion by mycobacterial deazaflavin-dependent nitroreductase followed by the release of intracellular NO, eradicating therefore a dormant *Mtb* subpopulation [86,87].

Knowledge of the rare cofactors and cofactor-dependent oxidoreductases accumulated in dormancy may contribute to the development of new redox compounds. Additionally, the bioactivation of prodrug compounds in hydrolytic processes, saved intact in dormancy is potent for the development of new anti-latent TB chemicals. Preliminary success of this concept has been demonstrated for nitazoxanide and methotrexate, albeit the particular bioactivating hydrolases were not disclosed [11]. Malfunctioning of the reactive carbonyl detoxification systems under a surplus of glycerol may lead to the accumulation of (methyl)glyoxal/lactaldehyde in dormant mycobacteria [55]. *In vivo*, a fat-rich diet may result in the accumulation of similar toxic carbonyl compounds within the cells: glyceraldehyde, glyceron-phosphate, methylglyoxal, crotonaldehyde, malondialdehyde, 4-hydroxynonenal and, therefore, the application of inhibitors of the carbonyl detoxifying process may lead to inactivation of the bacteria [88,89]. On the other hand, lipid catabolism may elicit 'ketone body'

intracellular acidosis (in the case of surplus acetyl-CoA) and malfunction, in particular, of malate synthase (*Rv1837c*) [90]. A possible malate synthase inhibitor, mimicking the structure of methyl glyoxal–phenyl-diketo acid inhibitor–was recently proposed [91] and its applicability for the inactivation of dormant mycobacteria looks promising.

## Conclusions

In conclusion, the current study demonstrated the biochemical adaptation of *Mtb* to the conditions of 'non-culturability' and visually demonstrated involvement of the pH-dependent regulator PhoP in the suppression of components of the DOS regulon and in the activation of a number of processes crucial for cellular survival. Further, biochemical processes essential for the development and support of dormancy (methylcitrate and glyoxylate pathways, the salvage of mycothiol and UMP biosynthetic pathways as well as protective systems) were revealed. In the present study we, for the first time, annotate enzymes significantly abundant in dormant *Mtb* cells (oxidoreductases and hydrolases), which potentially may be further considered as biotransformatory enzymes–prodrug activators suitable for the elimination of dormant mycobacteria.

## Supporting information

**S1 Table. The results of proteomic LC-MS analysis of *M. tuberculosis* H37Rv in dormancy and activity.** The supplementary table contains: the raw intensities of the detected and annotated proteins–"Raw LC-MS data"; the results of data $log_2$ transformation and quantile normalization–"Normalization" section; the results of differential analysis–"Results of differential analysis"; a reference list of antigenic proteins from published works on studies of the sera of human TB patients–"Antigenic proteins" sheet; a list of enzymes of dormancy as potential intracellular prodrugs activators "Enzymatic proteins" sheet.
(XLSX)

## Author Contributions

**Conceptualization:** Vadim Nikitushkin, Margarita Shleeva.

**Data curation:** Vadim Nikitushkin, Filip Dyčka F.

**Formal analysis:** Vadim Nikitushkin, Filip Dyčka F.

**Funding acquisition:** Margarita Shleeva, Filip Dyčka F.

**Investigation:** Dmitry Loginov, Filip Dyčka F.

**Methodology:** Margarita Shleeva, Jan Sterba, Arseny Kaprelyants.

**Project administration:** Margarita Shleeva, Jan Sterba.

**Supervision:** Margarita Shleeva, Jan Sterba, Arseny Kaprelyants.

**Validation:** Filip Dyčka F.

**Visualization:** Vadim Nikitushkin.

**Writing – original draft:** Vadim Nikitushkin.

**Writing – review & editing:** Vadim Nikitushkin, Margarita Shleeva, Dmitry Loginov, Filip Dyčka F., Jan Sterba, Arseny Kaprelyants.

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
