## [Decision Letter · Decision Letter 0]

22 Feb 2022

PONE-D-21-32365

Shotgun proteomic profiling of dormant, ‘non-culturable’ Mycobacterium tuberculosis

PLOS ONE

Dear Dr. Nikitushkin,

Thank you for submitting your manuscript to PLOS ONE. After careful consideration, we feel that it has merit but does not fully meet PLOS ONE’s publication criteria as it currently stands. Therefore, we invite you to submit a revised version of the manuscript that addresses the points raised during the review process.

ACADEMIC EDITOR: As recommended by the reviewers, minor modification is needed for this manuscript. Please use this opportunity to further improve the quality and visibility of your manuscript by thoroughly reviewing/editing for any errors.

We look forward to receiving your revised manuscript.

Kind regards,

Selvakumar Subbian, Ph.D.

Academic Editor

PLOS ONE

https://journals.plos.org/plosone/s/fileid=ba62/PLOSOne_formatting_sample_title_authors_affiliations.pdf".

“This work was funded by the Russian Science Foundation – Grant 19-15-00324 (microbiological experiments: cells cultivation and quantification, extracts preparation, data analysis and manuscript preeparation) and by the European Regional Development Fund-Project "Mechanisms and dynamics of macromolecular complexes: from single molecules to cells" (No. CZ.02.1.01/0.0/0.0/15_003/0000441) – LC-MS profiling of cells’ extracts and proteins annotation.”

“This work was funded by the Russian Science Foundation – Grant 19-15-00324 (microbiological experiments: cells cultivation and quantification, extracts preparation, data analysis and manuscript preeparation) and by the European Regional Development Fund-Project "Mechanisms and dynamics of macromolecular complexes: from single molecules to cells" (No. CZ.02.1.01/0.0/0.0/15_003/0000441) – LC-MS profiling of cells’ extracts and proteins annotation.

Reviewers' comments:

Reviewer's Responses to Questions

**Comments to the Author**

1. Is the manuscript technically sound, and do the data support the conclusions?

Reviewer #1: Partly

Reviewer #2: Yes

2. Has the statistical analysis been performed appropriately and rigorously? 

Reviewer #1: Yes

Reviewer #2: Yes

3. Have the authors made all data underlying the findings in their manuscript fully available?

Reviewer #1: Yes

Reviewer #2: Yes

4. Is the manuscript presented in an intelligible fashion and written in standard English?

Reviewer #1: Yes

Reviewer #2: Yes

5. Review Comments to the Author

Reviewer #1: This work describes the proteins / antigen that are differentially regulated during dormancy. The study was carried out systematically and is more significant to enrich the current knowledge of the biochemical pathway and immunogenic altercations during Mtb dormancy. The authors have extensively generated knowledge in the post persistence state of the pathogen, however the following needs to be addressed.

1. Among the various hypothetical invitro models proposed by various researchers in this field related to non-replicating persisters, some of them were extensively studied for their closeness to invivo conditions through evidences that might be analogous to the chain of events occurring while the pathogen is proceeding to a state of persistence invivo. The authors need to provide such evidences for this particular model of self-acidification, so that consequential elucidation of the events leading to knowledge generated reveal the closeness to those events occurring invivo. It is difficult to find such cross references for this particular model which might be significant to readers to assimilate possible post latency mechanistic consequences described in the manuscript.

2. Time-point of CFU and MPN was not clearly specified in the methods section. Also state whether the absence of contamination was ensured periodically.

3. The study was carried out in laboratory strain. It would have been better if the same was done in the clinical strain with different treatment outcomes.

Reviewer #2: In this study, the authors have identified differential proteome profile between actively growing and dormant Mycobacterium tuberculosis. This is the continuation of their previous study where they found diverse protein production in non culturable cells. I found the study very informative and well presented. However, I would suggest to provide the information from section "3.3 Differential analysis" in the tabular form as well.

6. PLOS authors have the option to publish the peer review history of their article (what does this mean?). If published, this will include your full peer review and any attached files.

Reviewer #1: **Yes: **Azger Dusthackeer

Reviewer #2: No

---

## [Author Response · Author response to Decision Letter 0]

6 Apr 2022

Reviewer 1: 

“This work describes the proteins / antigen that are differentially regulated during dormancy. The study was carried out systematically and is more significant to enrich the current knowledge of the biochemical pathway and immunogenic altercations during Mtb dormancy. The authors have extensively generated knowledge in the post persistence state of the pathogen, however the following needs to be addressed.

1. Among the various hypothetical invitro models proposed by various researchers in this field related to non-replicating persisters, some of them were extensively studied for their closeness to in vivo conditions through evidences that might be analogous to the chain of events occurring while the pathogen is proceeding to a state of persistence invivo. The authors need to provide such evidences for this particular model of self-acidification, so that consequential elucidation of the events leading to knowledge generated reveal the closeness to those events occurring invivo. It is difficult to find such cross references for this particular model which might be significant to readers to assimilate possible post latency mechanistic consequences described in the manuscript.”

- We have added a detailed description in Introduction section why our model is applicable for modeling of latent TB in vivo. 

«2. Time-point of CFU and MPN was not clearly specified in the methods section. Also state whether the absence of contamination was ensured periodically.»

- Added, Lines 186 (section “Evaluation of cell viability by CFU and MPN assays”) & 342 ( section “Dormant mycobacterial cells subjected to the proteomic analysis and the results of proteomic profiling”) in the revised manuscript.

«3. The study was carried out in laboratory strain. It would have been better if the same was done in the clinical strain with different treatment outcomes.»

- We agree, it could be interesting to use a clinical strain. At the same time, the most important publications on the subject from different laboratories were conducted on the H37Rv strain - what makes its crucial for the comparison of the results obtained in the current study with the previously published results.

Reviewer 2: 

«In this study, the authors have identified differential proteome profile between actively growing and dormant Mycobacterium tuberculosis. This is the continuation of their previous study where they found diverse protein production in non culturable cells. I found the study very informative and well presented. However, I would suggest to provide the information from section "3.3 Differential analysis" in the tabular form as well.»

- The results of statistical significance can be found in the supplementary excel file: “S1. Table. The results of proteomic LC-MS analysis of M. tuberculosis H37Rv in dormancy and activity” in the datasheet “The results of differential analysis”. 

The corresponding indices were similarly added for the sections “Annotation of immunogenic proteins accumulated in dormant cells”

and “Annotation of proteins accumulated in dormancy – potential prodrug activators” as well – where the lists of antigenic and enzymatic proteins are discussed.

---

## [Editor Report · Decision Letter 1]

11 Apr 2022

PONE-D-21-32365R1Shotgun proteomic profiling of dormant, ‘non-culturable’ Mycobacterium tuberculosisPLOS ONE

Dear Dr. Nikitushkin,

Thank you for submitting your manuscript to PLOS ONE. After careful consideration, we feel that it has merit but does not fully meet PLOS ONE’s publication criteria as it currently stands. Therefore, we invite you to submit a revised version of the manuscript that addresses the points raised during the review process.

ACADEMIC EDITOR: The authors have addressed the reviewer comments satisfactorily. However, i would like the authors to pay attention to the representation of figures and statistical information. For example, Figures 1-3, the x- and y- axis description can be more informative than just writing "value". Wherever comparison was made between groups, show the significance on the graph with * symbol and indicate what groups were compared. Also show what are the differentially colored groups in these figures. These details are only partially indicated in the figure legend. Therefore, attention needs to be paid to improving the "readability" of the figures.  These efforts would make the figures more understandable and meaningful to the reader. Finally, please check if you have disclosed all the "in-house" statistical scripts in the supplementary information.

We look forward to receiving your revised manuscript.

Kind regards,

Selvakumar Subbian, Ph.D.

Academic Editor

PLOS ONE
---

## [Editor Report · Decision Letter 2]

30 May 2022

Shotgun proteomic profiling of dormant, ‘non-culturable’ Mycobacterium tuberculosis

PONE-D-21-32365R2

Dear Dr. Nikitushkin,

We’re pleased to inform you that your manuscript has been judged scientifically suitable for publication and will be formally accepted for publication once it meets all outstanding technical requirements.

Kind regards,

Selvakumar Subbian, Ph.D.

Academic Editor

PLOS ONE
---

## [Editor Report · Acceptance letter]

29 Jul 2022

PONE-D-21-32365R2 

Shotgun proteomic profiling of dormant, ‘non-culturable’ *Mycobacterium tuberculosis*

Dear Dr. Nikitushkin:

I'm pleased to inform you that your manuscript has been deemed suitable for publication in PLOS ONE. Congratulations! Your manuscript is now with our production department. 

Kind regards, 

on behalf of

Dr. Selvakumar Subbian 

Academic Editor

PLOS ONE